# Being Creative Makes You Happier: The Positive Effect of Creativity on Subjective Well-Being

**DOI:** 10.3390/ijerph18147244

**Published:** 2021-07-06

**Authors:** Cher-Yi Tan, Chun-Qian Chuah, Shwu-Ting Lee, Chee-Seng Tan

**Affiliations:** Department of Psychology and Counselling, Universiti Tunku Abdul Rahman (UTAR), Kampar 31900, Perak, D. R., Malaysia; tancheryi@ymail.com (C.-Y.T.); chunqianchuah@gmail.com (C.-Q.C.); ting5541@hotmail.com (S.-T.L.)

**Keywords:** creativity, employees, experiment, priming, subjective well-being

## Abstract

The impact of happiness on creativity is well-established. However, little is known about the effect of creativity on well-being. Two studies were thus conducted to examine the impact of creativity on subjective well-being. In the first study, 256 undergraduate students (Study 1a) and 291 working adults (Study 1b) self-reported their creativity, stress, and subjective well-being. Hierarchical multiple regression analysis showed a positive relationship between creativity and subjective well-being after controlling the effect of self-perceived stress and demographics in both samples. Study 2 then employed an experimental design to examine the causal relationship between creativity and subjective well-being. Half of the 68 undergraduates underwent a creativity priming task followed by a divergent thinking test as well as self-reported stress and subjective well-being. The priming task was found to boost creative performance in the pilot study (Study 2a) and the actual study (Study 2b). Moreover, after controlling the effect of self-perceived stress, ANCOVA analysis showed that participants receiving the priming reported higher subjective well-being scores than their counterparts in the control group. The overall findings not only shed light on the facilitative effect of creativity on subjective well-being but also highlight the necessity of considering the reciprocal relationship of the two constructs in future research.

Studies have found a bidirectional relationship between creativity and well-being. On the one hand, well-being was found to promote creativity [1,2,3,4], on the other hand, creativity is conducive to well-being [5,6,7]. Nevertheless, the latter has received relatively little attention. Therefore, this research intended to investigate the beneficial effect of creativity on subjective well-being in the Malaysian context for two reasons. First, most of the past studies focused on happiness, positive affect/mood, psychological well-being, emotional well-being, or a combination of them. It is noteworthy that these constructs are conceptually different from subjective well-being. Specifically, unlike the abovementioned proxies that focus on the positive dimension, subjective well-being focuses on both positive and negative aspects of life as well as satisfaction with life [8,9]. Hence, it is essential to examine if creativity also contributes to subjective well-being. In addition, studies found cultural differences in the perception of creativity [10,11]. For instance, while novelty and usefulness are equally important in the West, usefulness has received more attention than novelty in the East [10]. As most of the past findings were derived from Western populations, it is essentially good to know if the same findings can be observed in Asian populations.

## 1. Creativity

Creativity is generally conceptualized as the ability to create products that are original and adaptive [12]. Nevertheless, originality alone is insufficient. To be considered creative, the output must also be applicable and useful to the problems at hand [13]. Indeed, empirical evidence supports that both novelty and usefulness are the two key ingredients of creativity [14,15]. Creativity is also regarded as a continuum as asserted by the Four C Model of creativity [16] that creativity ranges from mini-c (i.e., personally meaningful creativity), little-c (i.e., everyday creativity), Pro-c (i.e., professional creativity), to Big-C (i.e., eminent creativity).

Thus far, a variety of measurements have been proposed to capture the dynamic characteristics of creativity. Some researchers have focused on observable outputs such as involvement in creative activities and past achievements [17]. In a similar vein, the divergent thinking (DT) test has been extensively used as a proxy of creativity. Respondents were asked to generate as many solutions as possible to a given problem [18]. The solutions were then scored for fluency, flexibility, originality, and elaboration [19,20]. A meta-analysis supported that DT is a good predictor of creative achievement [21].

Meanwhile, self-reports with the advantages of cost-efficient and easy administration were commonly employed [22,23,24]. Tan et al. [11] developed the short form of the Kaufman Domains of Creativity Scale [25] with 20 items to help respondents indicate their belief of creativity relative to their peers in five different domains (e.g., scholarly, artistic). A self-rated creativity scale was also used for individuals to report their creative behaviors. For instance, Reiter-Palmon et al. [26] modified the items that were first developed for supervisors to evaluate the creativity of their employees [27,28] and for university students to report their perceptions of creativity in different domains (e.g., work, hobby). For example, the item “the employee suggests new way to achieve goals or objectives” was modified to “I suggest new way to achieve goals or objectives.” Similarly, Tan and Teo [29] modified the 13 items developed by Zhou and George [28] for undergraduate students in Malaysia to self-report their creativity. The modified self-rating items demonstrated good psychometric properties in the Malaysian context [15].

Notably, the consensual assessment technique (CAT) [30] was recognized as the gold standard for assessing the creativity level of a generated output (e.g., idea, performance). The CAT requires a panel of experts or professional individuals in the field to evaluate a particular product with the rationale that experts in a field can better identify elements that have made a creative contribution to the field.

## 2. Subjective Well-Being

Diener [31] has defined subjective well-being as individuals’ evaluation of their lives. He proposed the tripartite model of subjective well-being which consists of positive emotions, negative emotions, and satisfaction with life. Positive emotions include pleasant, happy, and joyful feelings, while negative emotions involve unpleasant, angry, and sad feelings [9]. Life satisfaction refers to the satisfaction of an individual with every aspect of his/her life [3]. The key components of the tripartite model of subjective well-being are attributed to those who have used the theory in later studies [32,33]. Accordingly, individuals with high levels of subjective well-being tend to have more positive emotions, less negative emotions, and high life satisfaction [34].

Numerous studies have shown positive effects of subjective well-being on physiological health. For instance, literature has revealed that happy individuals have better self-rated physical health [35]. In addition, subjective well-being is also linked with a reduced likelihood of all-cause morbidity among the older population [36]. Besides being one of the protective factors of cardiovascular disease [37,38] and incident cardiometabolic conditions [39], subjective well-being is also linked with positive outcomes in the workplace such as work-related productivity and success [40,41], job satisfaction [42], and professional performance and low absenteeism [43]. Taken together, having a high level of subjective well-being is important for both physical health and workplace performance. Other than that, well-being is also linked with one of the important mental abilities—creativity.

## 3. Creativity and Well-Being

The literature suggests a bidirectional relationship between creativity and well-being whereby positive affect, one of the components of well-being, was found to be related to creativity [1,2,3,4,44]. According to the broaden-and-build theory [45], positive states of moods such as joy and hope motivate people to explore and accept new information which can improve the flexibility of cognition and creativity [46,47]. People feel safe and secure when they are in positive emotions, and hence, are more likely to think divergently without fear [48]. Moreover, people are more open to changes when they are in positive moods [49]. On the contrary, negative emotions lead to people feeling insecure and view their environment as threatening. Therefore, individuals tend to use a more effortful and detail-oriented information processing style to deal with the situation [2]. Consequently, negative emotions reduce the willingness to take a risk and make unusual associations that impair their performance in creative problem-solving tasks [50]. Supporting the theoretical justification, self-report of creativity and (positive and negative) emotions collected from 658 undergraduate students using the Internet daily diary method across 13 days indicated that the participants reported higher creativity on days with higher levels of positive emotions [51]. In addition, literature also suggests that life satisfaction may strengthen individual effort and creativity. Specifically, individuals who are more satisfied with their lives are more willing to invest more effort in completing their jobs and making changes to solve problems [52].

Conversely, creativity was found to play a constructive role in well-being [53,54,55]. Chermahini and Hommel [56] randomly assigned university students to one of the four creativity tasks (i.e., divergent thinking, preparation for divergent thinking, convergent thinking, and preparation for convergent thinking). All participants reported their moods before and after working on the creativity task. Results showed that participants in the divergent thinking group reported a significantly higher score in positive mood than the other groups. Silvia et al. [57] lent further supports to the positive relationship between creativity and well-being using the experience-sampling method. Seventy-nine university students self-reported their emotions and involvement in creative activities (yes vs. no) for 7 days through a phone-based survey. The researchers found that participants were happier and more active when they were performing something creative. Similarly, supervisor-rated creativity was found to have a positive relationship with positive affect reported by employees (Study 1) and that employee-rated creativity reported at Time 1 predicted positive effect at work measured three months after Time 1 (Study 2) [58].

Although the abovementioned studies offer supports to the advantageous effect of creativity on positive affect, it is remarkable that the advantageous effect of creativity is not limited to the positive effect but the overall well-being. Puig et al. [59], for instance, found that breast cancer patients who received creative arts therapy intervention reported an improvement in the sense of psychological well-being. One of the reasons is the patients see their breast cancer experience as an opportunity for personal transformation and growth. In the same vein, individuals with creative occupations (e.g., town planners, architects, graphic designers) showed higher levels of well-being (e.g., life satisfaction, happiness) compared to those with non-creative professions (e.g., banker, insurance agent, accountant) [60]. Likewise, university students in Iran who reported high creativity and self-efficacy also reported high subjective well-being indexed by emotional, psychological, and social well-being [6]. The findings are consistent with the speculation of the flow theory [61]. Specifically, engaging in creative activities (e.g., composing a song, writing a script) allows individuals to enter flow states which are conducive to one’s life satisfaction and psychological well-being [62].

## 4. Overview of the Research

Building on the previous findings that creativity is positively associated with different proxies of well-being (e.g., happiness, positive mood, psychological well-being), the present research aimed to clarify the impact of creativity on subjective well-being. To be consistent with the focus on individual (perceived) well-being, we concentrated on mini-c and little-c in the present study.

Two studies were conducted to fulfill the objectives. Using a cross-sectional design, Study 1 examined the relationship between self-reported creativity (i.e., mini-c) and subjective well-being among undergraduate students (Study 1a) and working adults (Study 1b) respectively using hierarchical multiple regression. Meanwhile, Study 2 employed an experimental design to clarify the causal relationship between creativity evaluated by others (i.e., little-c) and subjective well-being using Analysis of covariate (ANCOVA). Unlike Study 1, creativity was induced in Study 2 by a priming task. The effectiveness of the priming task was tested and confirmed in a pilot study (Study 2a) and the main study (Study 2b) respectively using a divergent thinking task. All analyses were conducted using IBM SPSS Statistics 22. Ethical approval was obtained from the Scientific and Ethical Review Committee of Universiti Tunku Abdul Rahman (Ref no: U/SERC/92/2016). Informed consent was obtained from all participants.

The results are expected to offer empirical evidence to the beneficial cross-cultural effect of creativity on subjective well-being. Furthermore, the present study goes beyond past findings [56] to clarify the effect of creativity on subjective well-being by measuring and statistically controlling (perceived) stress. There are two merits to do so. First, as stress is detrimental to subjective well-being [63,64], stress may hinder the beneficial effect of creativity on subjective well-being. Therefore, controlling for stress will exclude the possible confounding effect of stress on the relationship between creativity and subjective well-being. Second, if creativity continues to show a positive impact on subjective well-being even after controlling for stress, the result implies that creativity contributes to subjective well-being directly. On the other hand, if the relationship between creativity and subjective well-being shifts from statistically significant to insignificant after controlling for stress, one of the possibilities is that stress could be a potential mediator. In other words, creativity benefits subjective well-being through stress reduction.

## 5. Study 1: Cross-Sectional Study

### Participants and Procedure

A total of 256 undergraduate students (Study 1a) and 291 working adults (Study 1b) in Malaysia participated in Study 1. Study 1a consisted of 154 females, while Study 1b consisted of 187 females. Participants aged 18 to 54 years old (*M* = 21.76, *SD* = 2.53) participated in Study 1a and those from 19 to 64 years old (*M* = 35.20, *SD* = 9.96) joined Study 1b. Most of the participants in both samples identified themselves as Chinese (81.9%), followed by Indians (8.8%), Malays (8.4%), and others (0.9%). In terms of religion, most of the participants were Buddhists, Taoists or Confucianists (70%), followed by Christians or Protestants (11.70%), Muslims (8.227%), Hindus (6.947%), and Atheists (3.108%). The inclusion criteria for Study 1a were being 18 years old and above and is currently enrolled in a bacherlor’s degree program, while the inclusion criteria for study 1b were being 18 years old and above and is currently employed or self-employed.

Participants from Study 1a and 1b were recruited by in-person method (e.g., researchers advertised the study to undergraduate students by presenting a poster in class) and online method (e.g., posting the recruitment advertisement on social networking sites such as Facebook and Twitter). All participants were screened to make sure they fulfill all inclusion criteria. Then, participants were briefed on the research objectives and procedure either by the researchers or by reading the information sheet on the first page of the online survey. Participants who gave their consent completed the survey in hardcopy or online.

## 6. Measurements

### 6.1. Subjective Well-Being

Self-reported subjective well-being was measured by the Scale of Positive and Negative Experience (SPANE) [9] and Satisfaction with Life Scale (SWLS) [8]. The SPANE requires participants to report their positive and negative feelings during the past week on 12 items (six items assessing positive and negative feelings). Each item was rated on a 5-point Likert scale ranging from 1 (*never*) to 5 (*always*). The positive and negative scales were scored separately where the total positive score (SPANE-P) and negative score (SPANE-N) ranged from 6 to 30, respectively. Additionally, affect balance scores (SPANE-B) were computed by subtracting the negative score from the positive score with the score ranging from −24 to 24 [9]. The SPANE was found to have good psychometric properties in a past study [65]: Cronbach’s coefficient ranged from 0.89 to 0.90 for SPANE-P, 0.84 for SPANE-N, and 0.88 for SPANE-B. SPANE is applicable for working adults and undergraduate students [9]. On the other hand, the SWLS is a 5-item scale designed to measure global cognitive judgments of one’s life satisfaction. Participants were asked to respond to each item on a 7-point Likert scale ranging from 1 (strongly disagree) to 7 (strongly agree). One of the items is “In most ways my life is close to my ideal”. Higher total scores would indicate more satisfaction with life and vice versa. Overall, the SWLS showed good internal consistency (α = 0.87 and the 2-month test-retest reliability was 0.82) and validity [66]. According to Li et al. [67], the total score of subjective well-being is derived as SWLS (life satisfaction) + SPANE-P (positive feeling)—SPANE N (negative feeling).

### 6.2. Creativity

Following Reiter-Palmon and colleagues’ [26] practice, we used the 15-item self-perceptions of creativity scale for participants to indicate their perceptions of creative ability using a 5-point Likert scale (1 = strongly disagree, 5 = strongly agree). By summing up all the item scores, higher total scores would indicate higher levels of creativity. The scale reported Cronbach’s alpha of 0.91 for its internal consistency [26]. A sample item was “*I utilize creative ideas that might improve conditions in my life*”.

### 6.3. Controlled Variable

As studies on different populations have consistently found that stress is negatively associated with happiness [68,69] and subjective well-being [63,64] stress was included as a covariate variable in the present study to exclude its confounding effect on the relationship between creativity and subjective well-being. The 10-item Perceived Stress Scale (PSS) [70] was thus employed for participants to self-report their stress levels in the past week using a 5-point Likert scale (0 = never, 4 = very often). A total score is obtained by reversing the scores of the four positive items (items 4, 5, 7, and 8) and then summing up all the item scores. The higher (total) ratings would indicate the higher levels of perceived stress [70].

## 7. Results and Discussion

### 7.1. Study 1a: Undergraduate Students

Table 1 shows descriptive statistics, correlation, and Cronbach alpha coefficient for each tested variable. Following Kim’s [71] suggestion for medium-sized samples (50 < *n* < 300), the normality assumption was met as the z values of skewness (i.e., skewness divided by standard error) and kurtosis were within 3.29. Pearson correlation analysis showed that the relationship between subjective well-being and self-perceived creativity was positive and statistically significant. Furthermore, a significant negative relationship was observed between subjective well-being and self-perceived stress, as well as between self-perceived creativity and self-perceived stress.

A two-step hierarchical multiple regression was conducted with subjective well-being as the dependent variable. Gender (male as reference), race (Chinese as reference), religion (Christian as reference), and self-perceived stress were entered at the first step (i.e., Model 1), while self-perceived creativity was entered at the second step (i.e., Model 2). The result showed that Model 1 was statistically significant, *F* (8, 247) = 21.65, *p* < 0.001 and accounted for 41.2% (adjusted R-square = 0.393) of the variance of subjective well-being. However, among the variables, only self-perceived stress had a significant relationship with subjective well-being, (standardized coefficient) β = −0.64, *SE* = 0.099, *t* = 12.84, *p* < 0.001, 95% confidence interval (CI) [−1.46, −1.07]. Similarly, Model 2 was also significant, *F* (9, 246) = 21.93, *p* < 0.001, and explained 44.5% of variance (adjusted R^2^ = 0.425), ∆*R*^2^ = 0.033, ∆*F* (1, 246) = 14.61, *p* < 0.001. Notably, after controlling the effect of demographic and self-perceived stress, self-perceived creativity displayed a positive relationship with subjective well-being, β = 0.20, *SE* = 0.068, *t* = 3.82, *p* < 0.001, 95% CI [0.13, 0.39].

### 7.2. Study 1b: Working Adults

Similarly, all the scales were found to have good internal consistency with target variables being normally distributed (see Table 1). There was a significant positive relationship between subjective well-being and self-perceived creativity. Meanwhile, subjective well-being and self-perceived creativity were negatively associated with self-perceived stress, respectively.

Model 1 was found statistically significant, *F* (9, 281) = 22.39, *p* < 0.001 and accounted for 41.8% (adjusted R^2^ = 0.399) of the variance of subjective well-being. Among the variables, self-perceived stress had a significant relationship with subjective well-being, β = −0.64, *SE* = 0.093, *t* = 0.13.72, *p* < 0.001, 95% CI [−1.45, −1.09]. Model 2 was also significant, *F* (10, 280) = 25.11, *p* < 0.001, and accounted for 47.3% (adjusted R^2^ = 0.454) of variance, ∆*R*^2^ = 0.055, ∆*F* (1, 280) = 29.27, *p* < 0.001. Self-perceived creativity was found to have a positive relationship with subjective well-being after controlling for demographic and self-perceived stress, β = 0.25, *SE* = 0.069, *t* = 5.41, *p* < 0.001, 95% CI [0.24, 0.51].

### 7.3. Discussion

Overall, the two studies consistently showed that self-rated creativity has a positive relationship with subjective well-being, even after statistically controlling the impact of self-perceived stress. The congruence not only offers preliminary support to the hypothesis but also indicates that the positive relationship between creativity and SWB applies to both young adults and working adults. Nevertheless, the results derived from such a cross-sectional design are unable to identify the causal relationship and are prone to common method bias as well as social desirability of maintaining a positive image. In the same vein, the design does not allow us to clarify the confounding effect of individual differences on the results. There is a possibility that the participants of the two studies have a higher level of creativity than the general. Thus, Study 2 was carried out to address the above-mentioned limitations and shed light on the causal relationship between creativity and subjective well-being using experimental design.

## 8. Study 2: Experimental Study

Study 2 aimed to investigate the causal relationship between creativity and subjective well-being using an experimental design. Creative performance was manipulated using a creativity priming task in which participants were instructed to describe their past creative behaviors. A pilot study (Study 2a) was first conducted to examine the effectiveness of the priming task, followed by the main study (Study 2b).

## 9. Study 2a: Validation of the Creativity Priming Task

### 9.1. Participants and Procedure

A minimum of 30 participants is required for conducting a pilot study [72]. We successfully recruited 48 undergraduates (38 females, aged 19 to 28, *M_age_* = 22.23, *SD* = 1.60) in this study. The majority of the sample were Chinese and final year students.

The computer-based experiment was conducted in a group of two to five participants in a laboratory. Participants were told that it was a study on problem-solving skills and well-being. After giving their consent, half of the participants were randomly assigned to the experimental group and the other half to the control group. Participants in the experimental group underwent a priming task followed by a creativity task (both tasks were explained in the Instruments section). Meanwhile, those in the control group did not undergo the priming task but only the creativity task. After completing the creativity task, the researchers debriefed and thanked the participants. A total of 10 sessions were conducted and each session took about 15 min.

### 9.2. Instruments

Creativity priming task. Being creative requires the ability to inhibit the activation of conventional routes of thinking [73]. However, individuals are less likely to intentionally suppress the tendency to think conventionally [74]. For example, although participants were explicitly instructed to generate new and original ideas, they were still unconsciously copied some features of the samples provided and produced uncreative ideas [75]. While modifying thinking style intentionally is complicated, cognitive processes can be influenced by priming to get around the predominantly activated knowledge and memory. For instance, Sassenberg et al. [76] showed that the memory recalling method is conducive to creative performance. Compared to their counterparts who intentionally tried to be creative, participants who were reminded of their earlier creative behaviors overcame inadvertent plagiarism and produced more creative ideas. The facilitative effect of the memory recalling method could be due to sample answers were not provided and thereby ruling out the impact of inadvertent plagiarism effect on the creativity task.

Given its effectiveness and advantages, the memory recalling method was employed in the present study. Our creativity priming task required participants to “briefly describe three situations in which they had behaved creatively” [74] in general in 5 min using English, Bahasa Malaysia, or Mandarin to minimize language barriers. Past studies have shown that the priming task is effective in stimulating a creative mindset [77] through the production of diverse, unique, and remotely associated ideas [74]. Such remembering and recalling creative activities involve transferring knowledge and information from long-term memory to working memory [78] and imaginatively reconstructing memories is related to creativity [79]. Moreover, the priming task is superior to the explicit instruction of being creative (with provided examples) as the induced creative mindset leads individuals to think differently and discard the influences of the given examples.

Creativity Task. The Guilford’s Alternative Uses Test (AUT) [80] was used to examine the effectiveness of the priming task. Participants were asked to complete the task individually by typing in the uses of stone(s) or rock(s) as many as possible in 5 min using their preferred language. Specifically, participants received the following instruction:

Please list the uses of stone/rock as many as possible. You are allowed to provide your answers in English, Bahasa Malaysia or Mandarin. You will automatically proceed to the next question after 5 min.

The responses were evaluated for originality, fluency, flexibility, and elaboration [80]. To assess originality, any responses that were given by only 1% of the participants would receive two points, while responses that were given by more than 5% of the participants received one point. In Study 2a, however, two marks were given to responses mentioned by one participant because 1% of the sample size (i.e., 48 participants) is less than 1. Fluency was represented by the total number of (non-repeated and sensible) answers given. Flexibility was evaluated by the number of categories that the participants used. For example, garden decoration and house decoration were considered as the same category, while house or building and fire starter were considered as two different categories. Finally, elaboration was determined by the number of details provided. Responses elaborated with functions or examples received one point, while two points were given for any further explanations [81]. Such priming manipulation is considered effective if the participants in the experimental group scored higher than those in the control group in the AUT.

In the study, the first three authors evaluated the participants’ responses to the AUT. Two of the raters were unaware of the grouping of participants and participants were coded numerically. All the raters share a similar cultural background with the participants. Prior to the commencement of the scoring process, the raters were asked to go through the responses to eliminate the repeated and non-sensical answers respectively. As the scoring method is rather objective (e.g., computing the total number of responses), the raters did not receive a particular definition of creativity.

Intraclass Correlation Coefficient (ICC) was used to measure inter-rater reliability to examine the impacts of individual differences on the rating scores. The result showed high consistency of the three raters in the evaluation results: ICC = 0.996, 95% CI [0.994, 0.998], *p* < 0.001 for originality, ICC = 0.998, 95% CI [0.997, 0.999], *p* < 0.001 for fluency, ICC = 0.986, 95% CI [0.977, 0.991], *p* < 0.001 for flexibility, and ICC = 0.984, 95% CI [0.973, 0.991], *p* < 0.001 for elaboration. Therefore, a mean score for each of the four creativity components was obtained by averaging the scores of three raters. Moreover, Pearson correlation analysis showed that the four creativity components significantly and positively correlated with each other: 0.329 (the relationships of fluency with elaboration and originality) < *r* < 0.884 (between fluency and flexibility). Hence, a composite creativity score was generated by summing up the four creative components’ mean scores to represent the creativity of participants.

## 10. Results

Manipulation checking. An independent-sample *t*-test was conducted to examine the effectiveness of the creativity priming task. The result showed a significant difference in the (composite) creativity score between the experimental group (*M* = 21.10, *SD* = 11.15) and the control group (*M* = 14.14, *SD* = 7.74), *t*(41) = 2.51, *p* = 0.03, Cohen’s *d* = 0.73. The results support that the creativity priming task is effective in promoting creative performance.

## 11. Study 2b: Testing the Effect of Creativity on Subjective Well-Being

### 11.1. Participants and Procedure

Another 68 undergraduates (50% female, aged 19 to 29 years old, *M_age_* = 21.97, *SD* = 1.41), who did not involve in Study 2a voluntarily participated in this experiment. The sample mainly comprised Chinese and final year students. The sample size was determined using G-Power analysis [82]. Using the following parameters: large effect size (Cohen’s f = 0.40), significant value (α) = 0.05, power (1—β) = 0.80, number of groups = 2, and number of covariates = 1, the recommended (total) sample size is 64.

After giving their consent, all participants answered the PSS. Then, half of the participants were randomly assigned to either the experimental group or the control group. The experimental group members underwent the creativity priming task first and followed by the AUT, whereas the control group members responded to the AUT directly without going through the priming task. Finally, all the participants answered the SWLS and SPANE at the end of the experiment. The experimental group spent 30 min while the control group spent 25 min in each session. A total of 14 sessions were conducted to collect data with all participants completing the tasks individually.

### 11.2. Instruments

The creativity priming task and AUT in Study 2a were used to induce creativity and test the effectiveness of the priming task. The responses of AUT were rated by the first three authors. As reported in Study 2a, the inter-rater reliability was good: ICC = 0.986, 95% CI [0.980, 0.991], *p* < 0.001 for originality, ICC = 0.985, 95% CI [0.978, 0.990], *p* < 0.001 for fluency, ICC = 0.979, 95% CI [0.968, 0.986], *p* < 0.001 for flexibility, ICC = 0.968, 95% CI [0.951, 0.979], *p* < 0.001 for elaboration, and the three raters’ scores were averaged. Moreover, a positive relationship was found among the four (mean scores of) creativity components: 0.473 (between originality and elaboration) < *r* < 0.908 (between fluency and flexibility) and therefore, the scores were summed to generate a composite creativity score.

Moreover, the SWLS (α = 0.83), SPANE (α = 0.90 for positive and 0.82 for negative), and PSS (α = 0.78) in Study 1 were included to assess participants’ subjective well-being and stress level.

### 11.3. Results and Discussion

Manipulation Checking. An independent-sample *t*-test was conducted to examine the effectiveness of the creativity priming task. There was a significant difference in the (composite) creativity score between the two groups: Levene’s test *F* = 5.61, *p* = 0.02, *t* (50.61) = 4.48, *p* < 0.001, *d* = 1.09. Specifically, the result (of equal variance not assumed) showed participants in the experimental group (*M* = 20.37, *SD* =11.06) outperformed their counterparts in the control group (*M* = 10.73, *SD* = 5.95). The same pattern was also observed on the four creativity components respectively. The statistical outputs were not presented here for the sake of clarity but are available upon request to the corresponding author. The consistent results further support that the priming task is advantageous in promoting creative performance.

Effect of Creativity on Subjective Well-being. An ANCOVA with group condition (creativity priming vs. no priming) as the independent variable, subjective well-being as the dependent variable, and stress as the covariate was conducted to examine the impact of creativity on subjective well-being. The homogeneity of variances assumption was not met, Levene’s test *F* (1, 66) = 12.29, *p* = 0.001. Results showed that self-rated stress had a significant effect on subjective well-being, *F* (1, 65) = 7.20, *p* = 0.009, $\eta_{p}^{\;2}$ = 0.100. More importantly, after controlling the effect of self-perceived stress, the effect of group condition was found significant, *F* (1, 65) = 5.75, *p* = 0.019, $\eta_{p}^{\;2}$ = 0.081. Specifically, the experimental group (*M* = 31.48, *SE* = 2.04) scored significantly higher than the control group (*M* = 24.55, *SE* = 2.04) in subjective well-being (*p* = 0.019).

Taken together, the findings indicated that creativity priming is effective in stimulating creative performance. Furthermore, participants who received the priming treatment reported a higher level of subjective well-being than their counterparts who did not receive any treatment. The results derived using an experimental study lend further support to the facilitative role of creativity in subjective well-being.

## 12. General Discussion

The relationship between creativity and well-being has been of interest to researchers. Both theoretical and empirical evidence has been established to the easing effect of well-being on creativity [1,4,48,49]. In contrast, the role of creativity in well-being received relatively less attention. Two studies were conducted in the present research to bridge the gap. The results support the hypothesized positive effect of creativity on subjective well-being.

Study 1 using a cross-sectional design not only investigated the positive relationship between creativity and subjective well-being among undergraduate students (Study 1a) but also examined the possibility of creativity in benefiting working adults’ subjective well-being (Study 1b). Both studies found that creativity is positively associated with subjective well-being.

The novelty of this research is the experimental examination of the impact of creativity on subjective well-being. Study 2b employed an experimental design and demonstrated the positive effect of creativity on subjective well-being. Participants in the experimental group had a higher level of subjective well-being compared to participants in the control group after controlling the effect of self-perceived stress. Such consistent results demonstrate that creativity promotes subjective well-being, and that the positive relationship applies to both young and working adults. Moreover, the result showed that the creativity priming task is effective in inducing one’s creative performance.

## 13. Implications of the Study

The present research has a few implications. Theoretically, the results contribute to the literature by shedding light on the relationship between creativity and subjective well-being. While most of the past studies focused on well-being that concentrates on the positive facet of life, our findings offer empirical evidence to support that creativity is positively associated with subjective well-being, a broader and more comprehensive measure of well-being.

In addition, the findings of the past studies [4] and present study jointly show that the relationship between creativity and subjective well-being is reciprocal rather than unidirectional. The results imply that well-being can enhance one’s creativity, and creativity can increase (subjective) well-being as well. Hence, practically speaking, individuals can improve their subjective well-being by fostering and strengthening their creativity. Furthermore, society may improve the publics’ mental health by promoting the beneficial role of creativity. Mental health professionals may even consider using creative activities to help their clients improve their subjective well-being. For example, art therapy was found useful in enhancing one’s subjective well-being [83]. Finally, the results lend further support to the flow theory [61] that being in a flow state through engaging in a divergent thinking task is conducive to one’s subjective well-being.

## 14. Limitations and Recommendations

There are nevertheless some limitations in the present research that deserve attention. First, the experimental and control groups in Study 2a and 2b had completed the experiment at different time span. Participants in the experimental group underwent the creative priming task before the AUT, while participants in the control group answered the AUT directly (i.e., without the priming task). Therefore, the difference in length of time may confound the performance of participants in the AUT. Future researchers are recommended to use an active control condition such as describing activities or ideas not related to creativity to further confirm that the found positive effect on subjective well-being is not merely because of recalling something. Note that, however, Sassenberg et al. [77] compared the creativity priming with preciseness priming (i.e., an active control) and no priming. They found that only creativity priming has a significant effect on creative performance and there was no significant difference between the preciseness priming and no priming conditions. Therefore, it is reasonable to believe that the effectiveness of the creativity priming observed in both Study 2a and 2b is plausible. Future researchers may also consider using a longitudinal design to verify the causal relationship between creativity and subjective well-being and shed light on the dynamic features (e.g., stability) of the relationship.

Furthermore, it is noteworthy that the DT test is a measure of creative potential which is different from actual creative behavior [84]. Moreover, the AUT responses were not evaluated by independent raters but the researchers. Although the inter-rater reliabilities were high in both studies, the results might be confounded by researcher biases. As a result, the effectiveness of the priming task may be overestimated. In that case, the priming task shall have no impact on both AUT and subjective well-being. Note that, however, participants who underwent the creativity priming reported higher creativity and subjective well-being scores than their counterparts in the control group as expected. The finding not only offers further support to the effectiveness of the priming task but also suggests that the impact of researcher bias is negligible if any. Nevertheless, it is still recommended to have external raters who are not aware of the research hypothesis and grouping of participants to eliminate the confounding effect of researcher biases.

Moreover, the present research did not pay attention to the underlying mechanism of the relationship between creativity and subjective well-being. Past studies found that creativity is a significant contributor to dishonest behaviors [85,86], and those dishonest behaviors can significantly predict subjective well-being [87]. Moreover, Tan et al. [69] demonstrated that creativity is indirectly associated with happiness through problem-solving ability and stress. These findings suggest that creativity could enhance subjective well-being indirectly via different paths. It is therefore a promising and crucial direction for researchers to explore the underlying mechanism of the effect of creativity on subjective well-being. In the same vein, the present research focused on subjective well-being (hedonic well-being). It is believed that engaging in creative tasks, especially tasks that are meaningful and congruent with one’s interest, is beneficial to functional (eudaimonic) well-being. This direction of study deserves future investigations.

Finally, the samples of the studies were mainly Chinese. It is risky and inadequate to generalize the facilitative effect of creativity on subjective well-being to the other ethnic groups in Malaysia. Future researchers are thus recommended to investigate the impact of creativity on different races and populations such as children and the elderly as well as other cultural groups in different countries.

## 15. Conclusions

Both creativity and well-being are essential to humankind. The present research not only replicates the positive relationship between creativity and well-being but also demonstrates that creativity is beneficial to subjective well-being. This new direction of study warrants more attention to expand the literature on positive psychology and creativity.

## Figures and Tables

**Table 1 ijerph-18-07244-t001:** Means, Standard Deviations, Correlation, and z values of Skewness and Kurtosis Among Variables for Study 1.

Variables	1	2	3	4	5	6
**Undergraduate Students**						
1.SWLS	(0.83)					
2.SPANE-Positive	0.40 ***	(0.84)				
3.SPANE-Negative	−0.30 ***	−0.44 ***	(0.87)			
4.SWB	0.79 ***	0.77 ***	−0.74 ***	-		
5.SPS	−0.37 ***	−0.49 ***	0.62 ***	−0.64 ***	(0.80)	
6.SPC	0.24 ***	0.33 ***	−0.12	0.30 ***	−0.19 **	(0.92)
*M*	21.30	22.27	17.15	26.43	19.44	52.01
*SD*	5.77	4.26	4.82	11.35	5.70	8.57
$Z_{Skewness}$	−0.83	0.02	1.26	−1.03	1.34	−0.63
$Z_{Kurtosis}$	−0.90	−2.02	−0.37	−0.10	1.37	2.56
**Working Adults**						
1.SWLS	(0.87)					
2.SPANE-Positive	0.31 ***	(0.85)				
3.SPANE-Negative	−0.31 ***	−0.37 ***	(0.87)			
4.SWB	0.78 ***	0.69 ***	−0.74 ***	-		
5.SPS	−0.39 ***	−0.47 ***	0.57 ***	−0.63 ***	(0.77)	
6.SPC	0.34 ***	0.35 ***	−0.23 ***	0.41 ***	−0.31 ***	(0.90)
*M*	23.35	22.19	15.19	30.35	17.91	54.20
*SD*	5.35	3.71	4.42	10.042	5.061	6.77
$Z_{Skewness}$	−4.02	1.48	2.10	−1.52	−2.81	0.64
$Z_{Kurtosis}$	1.67	−1.65	0.87	1.51	2.08	2.09

Note. SWLS = Satisfaction with Life Scale, SPANE = Scale of Positive and Negative Experience, SWB = subjective well-being, SPS = self-perceived stress, SPC = self-perceived creativity. ** *p* < 0.01, *** *p* < 0.001.

## Data Availability

The datasets generated during and/or analysis during the present study are available from the corresponding author on request.

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
