# Peer review of "Being Creative Makes You Happier: The Positive Effect of Creativity on Subjective Well-Being"

_ijerph, 2021, doi:10.3390/ijerph18147244_

Round 1

Reviewer 1 Report

The article has been improved especially by developing the limits of the study and building new directions of research.

Reviewer 2 Report

Thank you for allowing me to review this interesting study titled: Being Creative Makes You Happier: The Positive Effect of Creativity on Subjective Well-Being.

I consider that authors have addressed and revised all their manuscript according to my former comments. I am very grateful that some of my comments have been accepted and made explicit as limitations and future lines of research.

This manuscript is a resubmission of an earlier submission. The following is a list of the peer review reports and author responses from that submission.

Round 1

Reviewer 1 Report

The article contributes to the literature on the effects of creativity on well-being. As the authors surprised, there is a gap in the literature in this regard, while the inverse relationship (well-being promotes creativity) is demonstrated, if we mention only Barbara Fredrickson's theory. However, most studies that fall into the first category considered creativity in terms of mini-C. Indeed, there is also the article quoting the work of Fujiwara et al. (2015) made on a huge sample of individuals with creative occupations. However, the creativity included in Big-C can involve pleasure, satisfaction in certain stages of it but also involves torment.

The research is rigorously constructed in the four studies it proposes. Ingenious idea of ​​introducing creative priming task in the experiment;  and ingenious idea of introducing stress as covariate variable (covariate variable).

As minor issues:

- as it produced an example of an item in SWLS, it would be good to give an example of an item used in the instrument that measures creativity.

- it is important to write what program was used in the data analysis.

Reviewer 2 Report

The present work deals with a subject of great interest as it is the impact of creativity on subjective well-being. The methodology used is correct and, overall, the research design is acceptable.

I appreciate the authors’ efforts to design and develop this research.

After reading the manuscript, I have a few questions/suggestions that may help strengthen the paper further.

First, I think there should be more of a discussion of the development of creativity play in the study. Although participants self-reported their creativity, there aren´t other information or measurement in a study 1 (a,b) about their level of creativity. I don´t know if participants have high or low creativity in a baseline that could affect the global results. I mean, perhaps participants with high level of creativity in creativity tasks feels better subjective well-being than others less creative at a starting point of the research. 

On a related point, I think it would be helpful to have more detail about the development of creativity priming task from the study 2a (lines 332-342). On the other hand, the authors describe very well how the responses on creativity task were evaluated (352-353) and their results.

Third, with the methods, I’m curious about the choice to look the relation between creativity and subjective well-being from a static rather than a dynamic or developing viewpoint. Doing this type of research (longitudinal study) could make patterns in the relation between creativity and subjective well-being more readily apparent.

Overall, I think this is a strong paper, and I look forward to reading the final version.